# Theme Mapping and Bibliometrics Analysis of One Decade of Big Data Research in the Scopus Database

**Anne Parlina [1]** 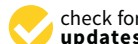**, Kalamullah Ramli [1,*]** **and Hendri Murfi [2]**

[1] Department of Electrical Engineering, Universitas Indonesia, Depok, Jawa Barat 16424, Indonesia; anne.parlina@ui.ac.id

[2] Department of Mathematics, Universitas Indonesia, Depok, Jawa Barat 16424, Indonesia; hendri@sci.ui.ac.id

[*] Correspondence: kalamullah.ramli@ui.ac.id; Tel.: +62-811-8606-457

**Abstract:** Recently, the popularity of big data as a research field has shown continuous and wide-scale growth. This study aims to capture the scientific structure and topic evolution of big data research using bibliometrics and text mining-based analysis methods. Bibliographic data of journal articles regarding big data published between 2009 to 2018 were collected from the Scopus database and analyzed. The results show a significant growth of publications since 2014. Furthermore, the findings of this study highlight the core journals, most cited articles, top productive authors, countries, and institutions. Secondly, a unique approach to identifying and analyzing major research themes in big data publications was proposed. Keywords were clustered, and each cluster was labeled as a theme. Moreover, the papers were divided into four sub-periods to observe the thematic evolution. The theme mapping reveals that research on big data is dominated by big data analytics, which covers methods, tools, supporting infrastructure, and applications. Other critical aspects of big data research are security and privacy. Social networks and the Internet of things are significant sources of big data, and the resources and services offered by cloud computing strongly support the management and processing of big data.

**Keywords:** bibliometrics; big data; clustering; science mapping; text mining

---

## 1. Introduction

Roger Mougalas from O'Reilly Media coined the term "big data" for the first time in 2005 [1]. It refers to a massive set of data that is hard to process and manage using traditional methods and tools. However, "big" as an adjective does not explain how large the data are. An explosion of data has been accumulated and processed in the last few decades, mainly comes from people's daily life and activities. Sophisticated information technology (IT) makes it easier to produce data [2]. Moreover, the connected world boosts big data. The development of advanced communication facilities and networks such as smartphones, the internet, social media, and others has caused the exponential growth of data volumes produced by people all over the world. Furthermore, the Internet of things (IoT) is another source of big data. Devices are being connected to the web and deliver data [3]. Data sets are proliferating, in part, because they are progressively being collected by inexpensive internet information-sensing devices and many things such as software logs, cellular devices, cameras, wireless sensor networks, remote sensing, and radio frequency identification (RFID) readers [4]. The International Data Corporation (IDC) predicts in a report that "the digital universe will grow from 2005 to 2020 by a factor of 300, from 130 exabytes to 40,000 exabytes, and until 2020 it will about double every two years [5]". Domo's seventh annual Data Never Sleeps infographic reveals that the world uses 4,416,720 gigabytes of internet data per minute [6].

The definition of big data itself has been much discussed by industry and academics. However, each speaker focuses on different aspects of big data, meaning that a consensus on the definition of big data is difficult to reach [7]. Big data itself is not new; it is being enabled by the advance of memory and storage technology, parallel and cloud computing, database technology, and innovative analysis tools. The IDC as a pioneer in studying big data, defines big data as follows: "Big data technologies describe a new generation of technologies and architectures, designed to economically extract value from huge volumes of a wide variety of data, by enabling high-velocity capture, discovery, and/or analysis" [8]. This definition describes four important characteristics of big data, namely volume, variety, velocity, and value. Ten years before, in a report from the META Group (now Gartner), Doug Laney pointed out that the opportunities and challenges of data growth have three dimensions; i.e., rising volume, velocity, and variety [9]. These three dimensions continue to be used to describe big data characters and are known as the "3 Vs". Volume refers to the vast amount of data that is generated all the time in the digital world. Velocity refers to the rapidity at which data are produced and the rapidity at which data move from one point to the next, whereas variety refers to increasingly diverse types or forms of data, such as images, text, networks, computer simulations, and geographical maps. Lately, more "Vs" was introduced to the big data community after challenges, and other ways of defining big data were discovered. In 2017, the number of "Vs" that had been raised had even reached 42 [10]. The term "big data" appears in many contexts [11], and the specific applications of big data can be found in almost any field of research [2].

The academic literature on big data has expanded in recent years. Thousands of scientific publications have been published regarding the theory, methods, infrastructure, and applications of big data technology. Extracting useful information and knowledge from these existing big data papers to identify the current developments and future trends in this field is a high-value research topic [12]. In the past, literature review coupled with expert opinions was the primary method for assessing the circumstances of technological developments. This method yields critical reviews that are often believed to capture the structure of knowledge and highlight the literature gaps [13]. However, experts tend to be biased and only focus on their respective fields of research, and analyzing an enormous number of papers can be a daunting task, even for an expert. Therefore, quantitative methods, such as bibliometrics and text mining, have been utilized. Bibliometrics applies statistical and mathematical methods to literature, which includes books, scientific, and technological publications, as well as other communication media [14]. It is a proven method for analyzing the relationship between literature or research stakeholders, and for evaluating research performance. However, it is still challenging to discover semantic connections between research topics from this point of view [15]. Advancements in natural language processing have led to the emergence of text mining as a tool for extracting useful patterns in unstructured data and analyzing semantic relationships in the scientific literature [16]. In this research, bibliometrics and text mining approaches were integrated to explore the research themes and obtain the research trends in the field of big data.

Some studies have attempted to uncover the hidden insights behind big data from existing scientific publications using various approaches such as surveys [17], bibliometrics analysis [12,18], social network analysis [19], systematic mapping [20], text mining [21], or a combination of these methods [22,23]. In addition to the studies above that discuss big data research trends in general, big data research trend analyses to reveal the role and influence of big data in a particular field are also performed, as explained in the following papers. Gu et al. explored the panorama of big data research in the field of healthcare informatics by conducting bibliometric analysis [24], while van Altena et al. applied the topic modeling approach to understand big data themes in the biomedical field [25]. Amado et al. performed a scientific literature analysis based on the text mining approach to identify the main trends in big data in marketing [26]. Shiddiqa et al. discussed big data management in terms of pre-processing, processing, security, and data storage [27]. Moreover, a study to identify the relations between companies and big data technologies using text mining and topic modeling method

was conducted by analyzing not the research publications but the news aggregated through Google News [28].

This paper aims to explore the worldwide research trends in the big data field and other relevant fields. The resulting information and knowledge may hold interest for students, academics, practitioners, science policy-makers, and R & D management in this field. The contributions of this study, relative to the recent publications in the field mentioned above, can be summarized as follows:

- We introduce a unique approach to exploring the major themes of text collection;
- We present the key-terms clusters analysis and capture the major themes in big data research;
- We provide a descriptive analysis of the big data research structure based on the growth of the number of publications, the most productive countries, institutions, and authors, and the most cited paper.

The remainder of this paper is organized as follows: Section 2 provides an overview of the big data system; Section 3 describes the data and methods of analysis used in the study; Section 4 presents the bibliometrics analysis; Section 5 explains the thematic analysis; Section 6 presents additional discussion; and Section 7 concludes the study.

## 2. An Overview of Big Data Systems

In principle, working with big data has the same basic requirements as working with any size of datasets. However, the design of big data systems must account for the unique characteristics of big data, which include massive scale (volume), speed of data collection and processing (velocity), and differences in data types, structure, and quality (veracity). Most big data systems aim to extract useful insights and patterns from massive volumes of heterogeneous data, which is not possible using conventional techniques [29]. To overcome the challenges of big data, a new generation of technology has emerged. Google pioneered many big data systems, including distributed file systems and MapReduce, which were responded to by the open-source community in subsequent years with the advent of Hadoop, Hive, HBase, Cassandra, MongoDB, RabbitMQ, and many more [30]. Figure 1 shows the components and general activities of a big data system.

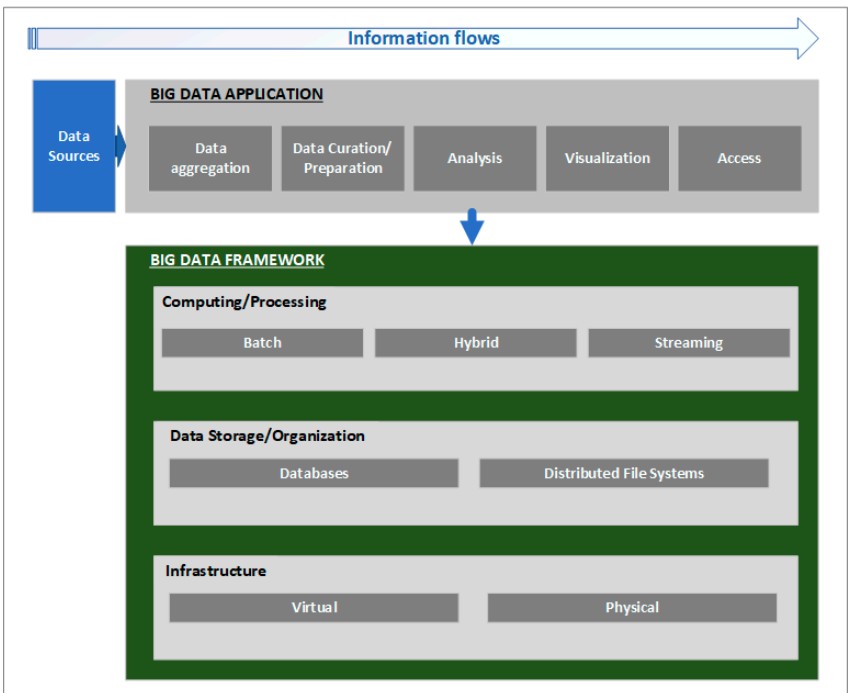

**Figure 1.** Big data systems overview.

## 3. Data and Methods

### 3.1. Data Analysis Framework

Figure 2 displays the framework for the analysis of the scientific publications in this study. It consists of three phases: data collection, data pre-processing, and data analysis. The experiments were conducted using python scripts for data pre-processing and T-LAB Plus—a text analysis software. T-LAB software is a set of linguistic, statistical, and graphical tools for the analysis of a text. It can be used in various text mining fields such as sentiment analysis, content analysis, thematic analysis, semantic analysis, text summarization, document clustering, etc. [31]. More detailed steps will be explained in the next section.

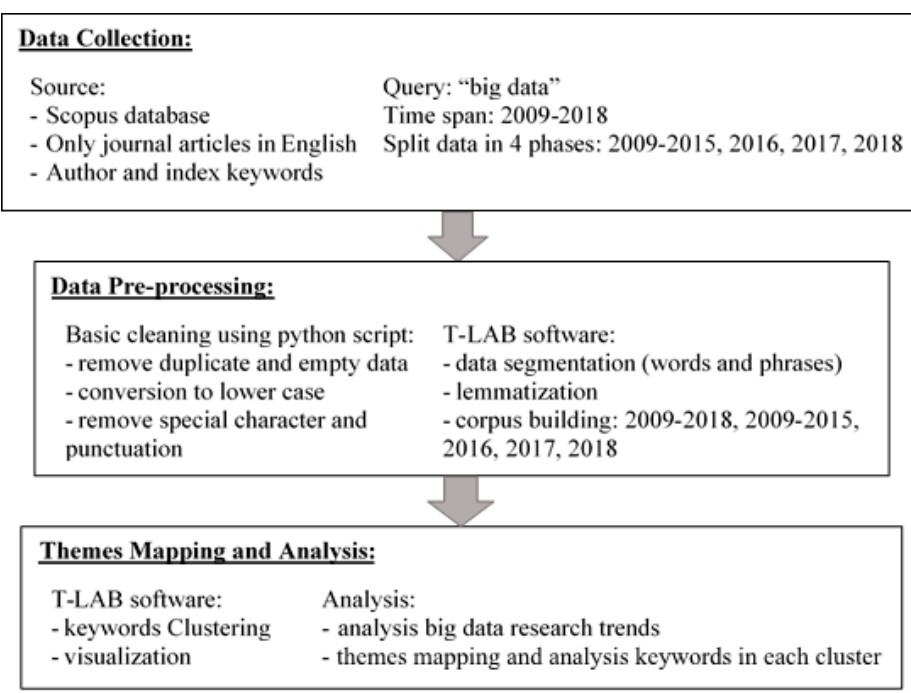

**Figure 2.** Analysis framework.

### 3.2. Data Collection

The first step of this study was collecting author and index keywords data of journal articles related to "big data" research. The Scopus database was chosen as a data source because it claimed to be the largest citation and abstract database of peer-reviewed scientific literature [32]. To obtain scientific publications related to this topic, the keyword "big data" was used as a search query in the Scopus database search interface. Scopus covers various types of documents, but only journal articles published in English and from the subject areas of Computer Science, Engineering, and Mathematics were included in this study. This study is limited to these three subjects to obtain a better understanding of big data technologies. The number of publications in these three fields is ca. 75% of the total number of publications regarding big data indexed in the Scopus database. Therefore, we believe that publications in these three fields are sufficient to represent big data research activities in the database. The data were retrieved on 1 August 2019.

Initially, 7321 journal articles published between 2009–2018 were collected. After the manual first screening, only 7274 articles were selected as the rest of the articles had duplicate or incomplete required data. To get a better insight into themes evolution, we divided the publications into four periods—2009–2015, 2016, 2017, 2018—with the number of documents in each period ranging between circa 1500–2300. Thus, there were five total analysis periods, as well as the total corpora that would be formed.

### 3.3. Data Pre-Processing

The pre-processing steps were intended to discard meaningless data and gain the relevant data from raw data records, which are used in corpora building for the text mining process. The corpora were composed of the papers' author and index keywords retrieved from bibliographic data records. All five corpora were prepared in two phases. Firstly, the raw data were pre-processed with basic natural language processing (NLP) tasks such as special character and punctuation removal and lowercase conversions. To obtain better results in terms extraction process, a controlled vocabulary was included in this process. A controlled vocabulary is an organized list of phrases and words for content indexing or retrieval. It captures the variation in terms and provides a consistent method to assign preferred terms for similar concepts [33]. Previous literature shows that integrating a controlled vocabulary into the extraction process of words and phrases can enhance the results [34–36]. We performed this phase using python scripts and delivered the data that were cleaned up in the csv file format. The next phase was to import these csv files into the T-Lab software corpus builder, in which the data were further processed. We utilized T-Lab features for removing stop words, lemmatization, and data segmentation into words and phrases.

### 3.4. General Analysis and Thematic Mapping

Bibliometrics is a set of methods used to measure and evaluate academic output [37]. There are two principal bibliometrics methods to evaluate and analyze a research field: performance analysis and science mapping. The purpose of performance analysis is to evaluate groups of research actors, i.e., researchers, countries, institutions (universities or departments), and the impact of their activity. Meanwhile, science mapping aims to extract the information and knowledge from the conceptual or social structure of the particular research field [38]. This study performs general analysis to depict the growth of the number of publications, the core journals, and the most productive countries, authors, and institutions. Furthermore, the theme evolution of big data research was explored by thematically clustering the elementary context in the corpora. In this case, elementary contexts were keywords and key-phrases.

Thematic mapping was conducted by thematically clustering the keywords and key-phrases using the bisecting k-means algorithm [39–41] in the T-Lab tool. The maximum number of clusters was set to 50, and the minimum co-occurrence of keywords within the context units was two. The clustering procedure in T-Lab software consists of the steps below:

a.  Construction of data vector (document's number x the occurrence frequency of keywords/key-phrases);
b.  Term frequency and inverse document frequency (TF-IDF) weighting [42] and scaling of row vectors to Euclidian norm;
c.  Bisecting k-means clustering with the cosine similarity measure.

The clustering results were refined using the naïve Bayes (NB) classifier method [43,44]. The cluster memberships of the i-keywords/key-phrases determined by the bisecting K-means must be the same as determined by the naïve Bayes classifier, and the maximum posterior value corresponding to the i-keywords/key-phrases must be at least 50% higher than its posterior value in other clusters. The keywords and key-phrases that do not meet the two criteria above will be deleted [45]. The following naïve Bayes classifier formula is used by T-Lab:

$$v_{NB} = \ \text{argmax} P\big(v_j\big) \prod_i P\big(a_i|v_j\big) \tag{1}$$

$$v_j \ \in \ V$$

where

| | |
|---|---|
| argmax | the maximum posterior value; |
| $v_j \in V$ | j-cluster ($v_j$) of the partion ($V$); |
| $P(v_j)$ | the prior probability of each j-cluster to their relative frequencies; |
| $\prod_i P(a_i\|v_j)$ | the probability product of each ($a_i$) term within each ($v_j$) cluster, where each |
| $P(a_i\|v_j)$ | is an element of a normalized vector of relative frequencies and each ($a_i$) term is present in the i-context unit to be reclassified. |

The relationships between clusters in bidimensional spaces can be visualized and explored by the graphs obtained from correspondence analysis [46]. The cluster labels can be proposed automatically by a specific T-Lab function.

## 4. Bibliometrics Analysis of Publication Trends

At first, the 7274 journal articles within the scope of big data were recapitulated corresponding to their publication years. The growth and distribution of the journal articles by publication years are depicted in Figure 3. Before 2012, only a few (three) journal articles were published. However, it can be seen from Figure 3 that the number of published journal articles about big data has obviously and continuously increased from 2012 to 2018. This indicates that progressively more attention has been paid to this research topic in recent years.

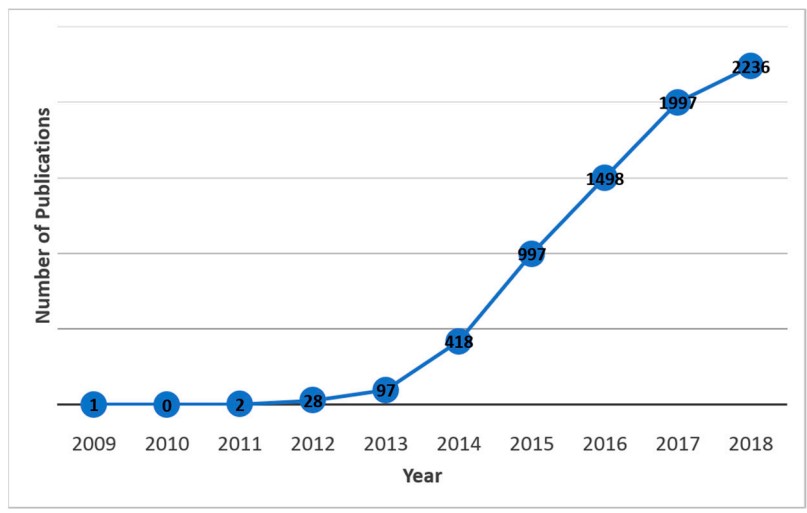

**Figure 3.** Journal article publication of big data research in the Scopus database per year between 2009–2018.

Figure 4 displays the top 10 journals in which the most papers in the big data field were published. A total number of 1144 articles regarding big data were published in these journals between 2013–2018 and indexed in the Scopus database. Before 2013, no journal used in this study were published in these ten journals. The top three journals were IEEE Access (278 papers), Future Generation Computer Systems (202 papers), and Boletin Tecnico Technical Bulletin (156 papers). Table 1 shows the name of journals, the total number of journal articles, and the number of articles per year published in each journal.

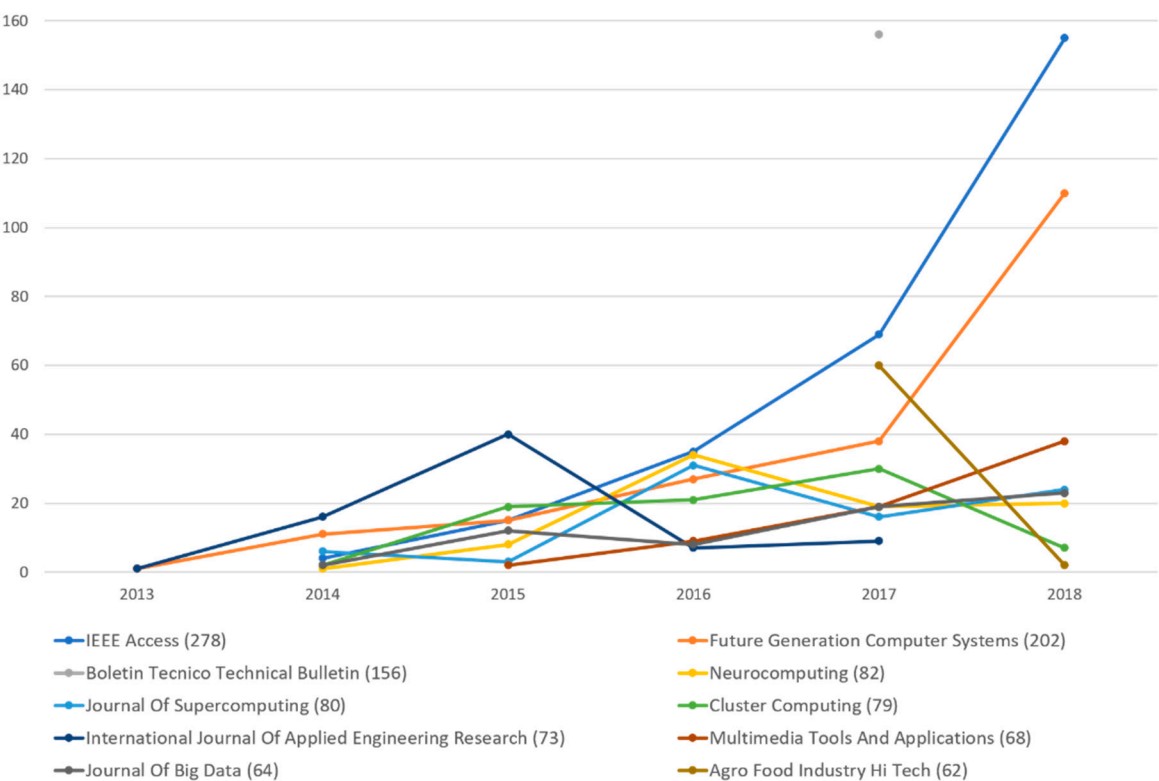

**Figure 4.** Top 10 journals in which the most numbers of documents regarding big data research were published.

**Table 1.** Top 10 journals, the number of documents regarding big data research yearly and total published.

| Journal | 2013 | 2014 | 2015 | 2016 | 2017 | 2018 | Total |
|---|---|---|---|---|---|---|---|
| IEEE Access | | 4 | 15 | 35 | 69 | 155 | 278 |
| Future Generation Computer Systems | 1 | 11 | 15 | 27 | 38 | 110 | 202 |
| Boletin Tecnico Technical Bulletin | | | | | 156 | | 156 |
| Neurocomputing | | 1 | 8 | 34 | 19 | 20 | 82 |
| Journal of Supercomputing | | 6 | 3 | 31 | 16 | 24 | 80 |
| Cluster Computing | | 2 | 19 | 21 | 30 | 7 | 79 |
| International Journal of Applied Engineering Research | 1 | 16 | 40 | 7 | 9 | | 73 |
| Multimedia Tools and Applications | | | 2 | 9 | 19 | 38 | 68 |
| Journal of Big Data | | 2 | 12 | 8 | 19 | 23 | 64 |
| Agro Food Industry Hi Tech | | | | | 60 | 2 | 62 |

A total of 104 countries have contributed to the publication of journal articles on this topic. Figure 5 shows the top 10 most productive countries in the big data research field according to the number of publications (Pub.). China had the highest number of publications, followed by the United States and the United Kingdom. More than one third (35.4%) of worldwide publications come from China. The total number of journal articles related to big data published by these three countries was more than half (66.97%) of world productivity. Table 2 summarizes the characteristics of big data publications published by the 10 most active countries. Other indicators, such as total citations (Cit.), Cit./Pub, H-index and the number of publications for each period are also presented to portray the characteristics of the top 10 productive countries in big data research. Besides, Table 2 displays the changes in total publications of these ten countries for each period over the past ten years.

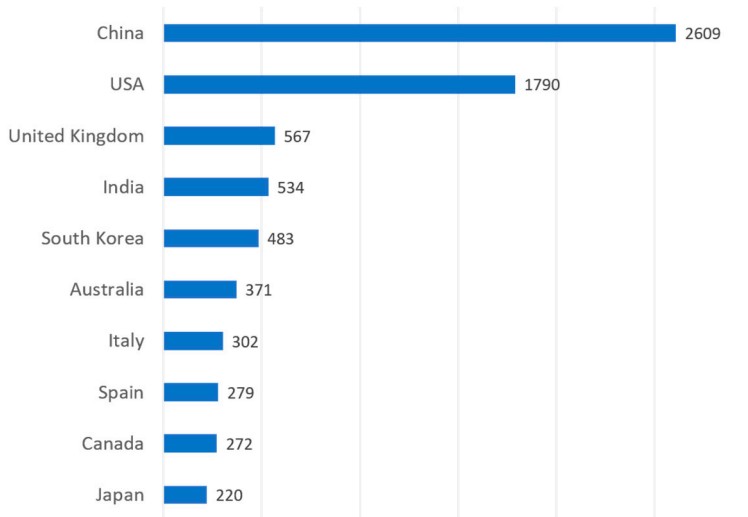

**Figure 5.** The top 10 productive countries in big data research.

**Table 2.** Top 10 productive countries in big data research.

| Country | H-index | Number of Publications (Pub.) | Total Citations (Cit.) | Cit./ Pub. | 2009–2015 | 2016 | 2017 | 2018 |
|---|---|---|---|---|---|---|---|---|
| China | 83 | 2609 | 21507 | 8.24 | 428 | 536 | 790 | 855 |
| USA | 88 | 1790 | 27171 | 15.18 | 530 | 369 | 395 | 496 |
| United Kingdom | 54 | 567 | 7386 | 13.03 | 123 | 127 | 133 | 184 |
| India | 33 | 534 | 1642 | 3.07 | 102 | 103 | 110 | 219 |
| South Korea | 32 | 483 | 3212 | 6.65 | 102 | 118 | 112 | 151 |
| Australia | 51 | 371 | 5607 | 15.11 | 88 | 68 | 95 | 120 |
| Italy | 31 | 302 | 2453 | 8.12 | 57 | 58 | 75 | 112 |
| Spain | 36 | 279 | 3141 | 11.26 | 55 | 63 | 63 | 98 |
| Canada | 42 | 272 | 4566 | 16.79 | 60 | 61 | 70 | 81 |
| Japan | 24 | 220 | 1150 | 5.23 | 70 | 49 | 40 | 61 |

Figure 6 reveals the top 10 most productive institutions in big data research publications according to Pub. The first ranked institution is the Chinese Academy of Sciences, which has published 190 articles in this research field. The second and third were Tsinghua University and the Ministry of Education China, which published 118 and 105 articles, respectively. The majority of these 10 institutions are from China (80%), and the rest are from India (10%) and Saudi Arabia (10%). Table 3 summarizes the characteristics of big data publications published by the 10 most active institutions. In addition to total publications (Pub.), Table 3 also shows other indicators (H-index, Cit., Cit./Pub.) to examine the impact and productivity of the top 10 institutions in the field of big data research. It can be seen from Table 3 that institutions from China dominate research activities in this field. In contrast, institutions from the USA, which ranked as the second most productive country, did not even appear on the list. A possible cause of this is the more even distribution of research on big data in the US, so there are no prominent institutions in this field like in China.

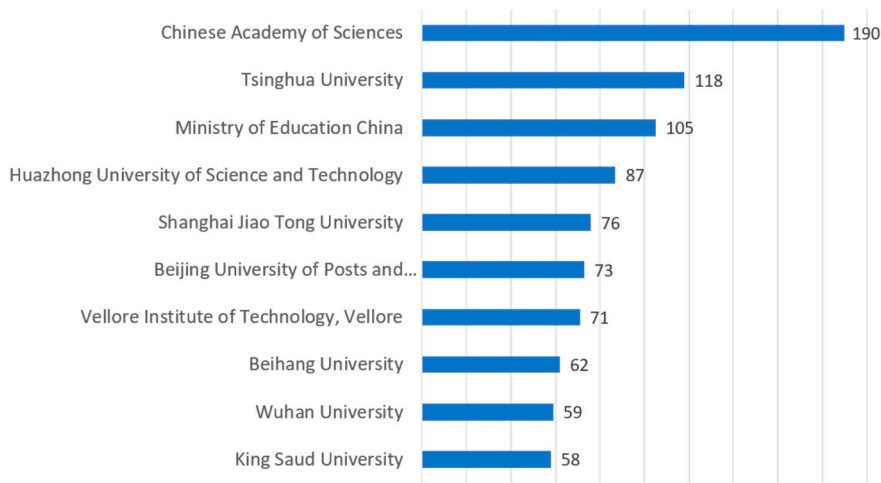

**Figure 6.** Top 10 most productive institutions in big data research.

**Table 3.** Top 10 productive Institutions in big data research.

| Institution | Country | H-index | Pub. | Cit. | Cit./Pub. | 2009–2015 | 2016 | 2017 | 2018 |
|---|---|---|---|---|---|---|---|---|---|
| Chinese Academy of Sciences | China | 31 | 190 | 2511 | 13.22 | 51 | 38 | 46 | 55 |
| Tsinghua University | China | 27 | 118 | 1667 | 14.13 | 35 | 29 | 24 | 30 |
| Ministry of Education China | China | 21 | 105 | 912 | 8.69 | 12 | 21 | 26 | 46 |
| Huazhong University of Science and Technology | China | 23 | 87 | 1161 | 13.34 | 13 | 19 | 21 | 34 |
| Shanghai Jiao Tong University | China | 19 | 76 | 722 | 9.50 | 10 | 18 | 27 | 21 |
| Beijing University of Posts and Telecommunications | China | 15 | 73 | 654 | 8.96 | 4 | 22 | 24 | 23 |
| Vellore Institute of Technology, Vellore | India | 15 | 71 | 288 | 4.06 | 8 | 21 | 11 | 31 |
| Beihang University | China | 19 | 62 | 739 | 11.92 | 7 | 13 | 20 | 22 |
| Wuhan University | China | 15 | 59 | 424 | 7.19 | 8 | 14 | 11 | 26 |
| King Saud University | Saudi Arabia | 21 | 58 | 870 | 15.00 | 1 | 10 | 18 | 29 |

The top 10 most productive authors are displayed in Figure 7. Three authors—Chang, V.; Yang, L.T.; and Zomaya, A.Y.—have the highest number of publications, namely 30 publications. These three authors are ranked first, while the second and third ranks are Ranjan, R. and Herrera, F., who published 26 and 24 journal articles respectively. Table 4 summarizes the number of articles and citations from the authors in four periods.

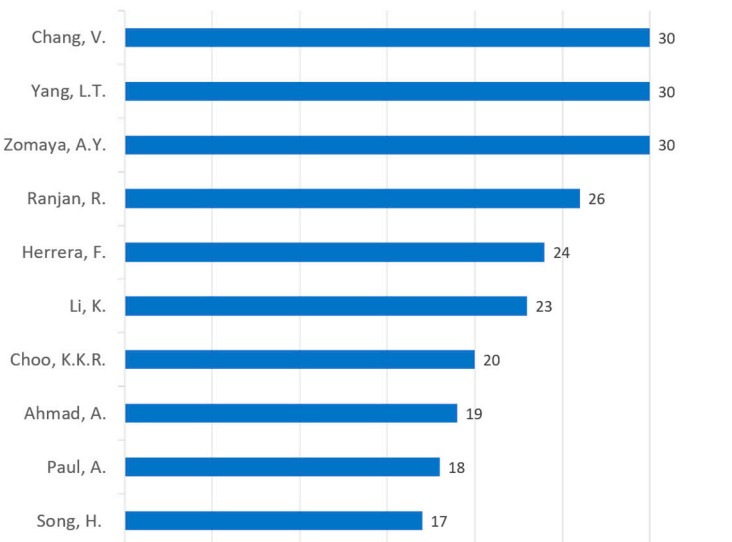

**Figure 7.** Top 10 most productive authors.

**Table 4.** The number of publications (Pub.) and citations (Cit.) of the top 10 most productive authors in four periods.

| | 2009–2015 | | 2016 | | 2017 | | 2018 | | Total | | |
|---|---|---|---|---|---|---|---|---|---|---|---|
| Author | Pub. | Cit. | Pub. | Cit. | Pub. | Cit. | Pub. | Cit. | Pub. | Cit. | Cit./Pub. |
| Chang, V. | 1 | 73 | 7 | 604 | 7 | 109 | 15 | 255 | 30 | 1041 | 34.7 |
| Yang, L.T. | 4 | 195 | 7 | 220 | 7 | 154 | 12 | 228 | 30 | 797 | 26.6 |
| Zomaya, A.Y. | 9 | 608 | 6 | 130 | 8 | 107 | 7 | 101 | 30 | 946 | 31.5 |
| Ranjan, R. | 12 | 696 | 2 | 34 | 5 | 95 | 7 | 82 | 26 | 907 | 34.8 |
| Herrera, F. | 8 | 801 | 2 | 99 | 7 | 170 | 7 | 129 | 24 | 1199 | 49.9 |
| Li, K. | 8 | 250 | 3 | 28 | 5 | 165 | 7 | 75 | 23 | 518 | 22.5 |
| Choo, K.K.R. | 1 | 60 | 3 | 71 | 8 | 215 | 8 | 63 | 20 | 409 | 20.4 |
| Ahmad, A. | 0 | 0 | 6 | 412 | 4 | 55 | 9 | 63 | 19 | 530 | 27.8 |
| Paul, A. | 0 | 0 | 6 | 412 | 4 | 55 | 8 | 81 | 18 | 548 | 30.4 |
| Song, H. | 1 | 35 | 7 | 380 | 4 | 154 | 5 | 76 | 17 | 645 | 37.9 |

The top 10 most cited journal articles by big data publications are presented in Table 5. The number of citations shows how popular the paper is among researchers in the big data area. Most of these papers are survey papers related to the theme of big data analytics, big data technology, and the internet of things (IoT). The first-ranked article, titled "Internet of Things: A Survey on Enabling Technologies, Protocols, and Applications", was authored by Al-Fuqaha et al. This paper presents an overview of the IoT concept, its enabling technologies, protocols, application problems, challenges, and issues, as well as the relationship between IoT and other emerging technologies such as big data analytics together with cloud and fog computing [47].

**Table 5.** Top 10 highly cited articles.

| No. | Title | Authors | Year | Cited by | Refs. |
|---|---|---|---|---|---|
| 1. | Internet of Things: A Survey on Enabling Technologies, Protocols, and Applications | Al-Fuqaha A. et al. | 2015 | 2072 | [47] |
| 2. | Business intelligence and analytics: From big data to big impact | Chen H. et al. | 2012 | 2058 | [48] |
| 3. | Data mining with big data | Wu X. et al. | 2014 | 1253 | [49] |
| 4. | Data-intensive applications, challenges, techniques, and technologies: A survey on Big Data | Philip Chen C.L. and Zhang C.-Y. | 2014 | 1202 | [50] |
| 5. | Beyond the hype: Big data concepts, methods, and analytics | Gandomi A. and Haider M. | 2015 | 993 | [51] |
| 6. | Traffic Flow Prediction with Big Data: A Deep Learning Approach | Lv Y. et al. | 2015 | 738 | [52] |
| 7. | The internet of things for health care: A comprehensive survey | Islam S.M.R. et al. | 2015 | 693 | [53] |
| 8. | Scalable nearest neighbour algorithms for high dimensional data | Muja M. and Lowe D.G. | 2014 | 606 | [54] |
| 9. | Big Data, new epistemologies and paradigm shifts | Kitchin R. | 2014 | 515 | [55] |
| 10. | Toward scalable systems for big data analytics: A technology tutorial | Hu H. et al. | 2014 | 475 | [7] |

## 5. Thematic Analysis of the Author and Index Keywords

In order to capture the theme map of big data research in detail, the author keywords and index keywords from bibliographic records were clustered and classified using bisecting k-means and naïve Bayes algorithms. Table 6 displays the total number of words and phrases (key terms), number, and the percentage of key-terms that were classified in the clustering process as explained above for five periods (2009–20018, 2009–2015, 2016, 2017, 2018). Figure 8 shows the number and percentage of key-terms classified in each cluster for the data period 2009–2018. The top 10 characteristics key-terms for each cluster are shown in Appendix A, while Figure 9 visualizes the relationships between the clusters. Each circle in the figure represents the key-terms cluster or topic. Appendix B shows the number and percentage of author and index keywords classified in each cluster, while Appendix C displays the cluster relationships for the four different periods (2009–2015, 2016, 2017, 2018). Appendices D–G present the key-terms in each theme.

**Table 6.** The number and percentage of key-terms classified in the thematic analysis process for each period.

| Period | Total Key-Terms | Number of Key-Terms Classified | Percentage |
|---|---|---|---|
| 2009–2018 | 9456 | 7086 | 74.94% |
| 2009–2015 | 1969 | 1612 | 81.87% |
| 2016 | 2045 | 1699 | 83.08% |
| 2017 | 2480 | 2032 | 81.94% |
| 2018 | 3064 | 2472 | 80.68% |

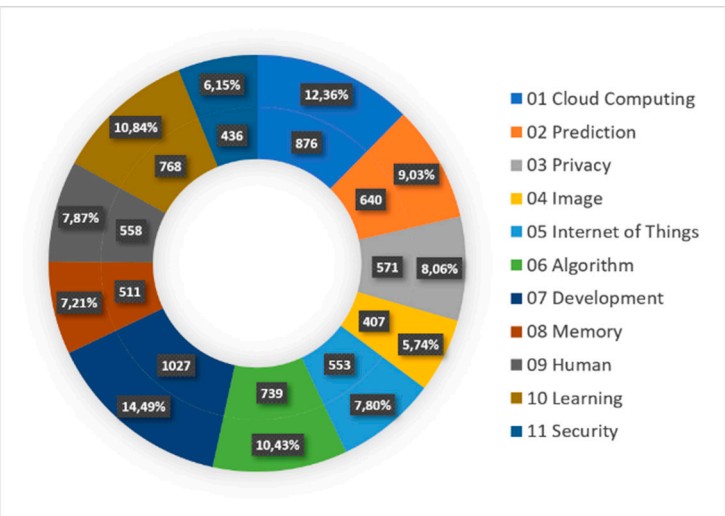

**Figure 8.** The number of key-terms in each cluster for the period 2009–2018.

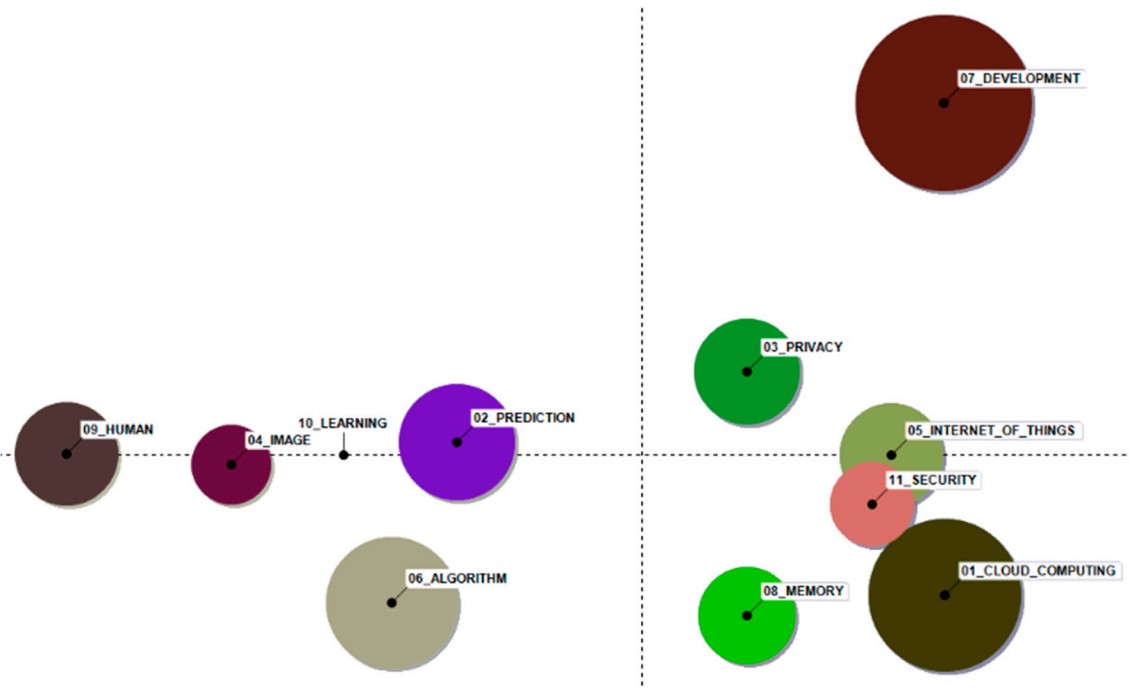

**Figure 9.** Key-terms clusters visualization (2009–2018).

*5.1. Major Themes in Big Data Research*

There are 11 key-terms clusters which resulted from the clustering process of the entire data period (2009–2018), whereas for the other four periods, the number of clusters is as follows: 2009–2015,

12 clusters; 2016, 13 clusters; 2017, 12 clusters; and 2018, 12 clusters. These key-terms clusters can be seen as topics or themes that dominate in big data research and can be further classified as follows.

### 5.1.1. Big Data Analytics Tools and Algorithms

Raw data do not have a meaning until they are processed and contextualized into useful information. Analytics is the process of extracting meaningful insight and information from raw data. There are four principal types of analytics: descriptive analytics (what has happened?), diagnostic analytics (why did it happen?), predictive analytics (what is likely to happen?), and prescriptive analytics (what can we do to make it happen?) [56]. Big data analytics involves data science, statistics, and machine learning. Some of the tools that are commonly used in data science are optimization, probability theory, linear algebra, graph, topology, visual analytics, programming languages and software, and other mathematic tools [57]. The National Research Council proposes the following seven tools of statistical data analysis for massive data sets that have proved to be useful in massive data analysis: basic statistics, generalized N-body problems, linear algebraic computations, graph-theoretic computations, optimization, integration, and alignment problems [58]. Meanwhile, there are several types of algorithms known in machine learning; among others, clustering, regression, classification, dimensionality reduction, frequent pattern mining, and feature extraction. Figure 10 displays some examples of machine learning algorithms.

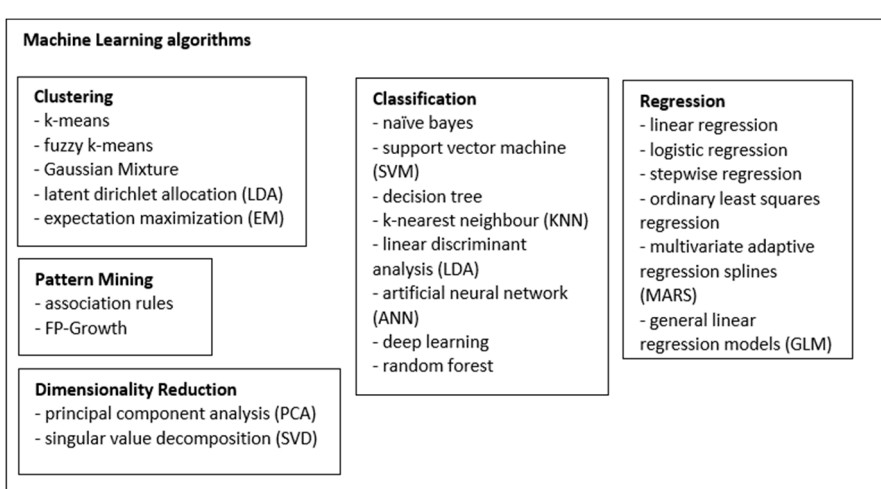

**Figure 10.** Some examples of machine learning algorithms.

Themes or topics related to big data analytics tools found in the results of key-terms clustering include prediction, machine learning, detection, classification, visualization, stochastic systems, data mining, optimization, and estimation.

### 5.1.2. Big Data Infrastructure

Big data infrastructure includes hardware, software, and network architecture that supports big data management and processing. It entails agents and tools that collect data, physical storage, and software systems that store the data, the network that transfers the data, the application environments that host the analytics tools, and the archive infrastructure that backs the data up [59]. Big data processing can be distinguished into two paradigms according to the processing time requirements: streaming processing and batch processing [7]. Figure 11 shows some components of big data architecture.

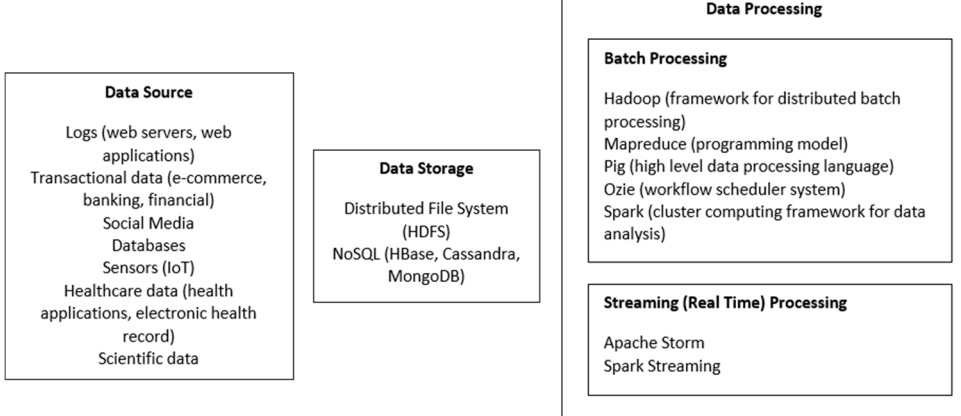

**Figure 11.** Some components of big data architecture.

The analysis of key-terms reveals the cluster's labels and key-terms related to big data technology; i.e., Hadoop, Map-reduce, Spark, NoSQL, memory, data compression, data storage, distributed computing, parallel computing, and computer systems.

### 5.1.3. Privacy and Security Issue in Big Data

Although big data offers great opportunities for business and the economy, exploring and using the extraordinary value of big data also increases security and privacy risks. Big data security refers to the implementation of solutions for increasing the security, reliability, and safety of big data technology. Meanwhile, big data privacy focuses on big data protection from unwanted inference and unauthorized use [60]. Without planned security and privacy solutions, big data could face big challenges and problems. Many facts indicate that big data endangers user privacy if not handled properly [61]. For instance, in the healthcare context, the exploitation of big data could save the healthcare industry up to US$ 450 billion. However, privacy issues become a principal consideration when exploiting big data in the healthcare field, as the data of patients are very sensitive [62]. On the other hand, security can take advantage of using an increasing amount of available data that can be analyzed in real-time for security applications such as attack classification and detection [60].

### 5.1.4. Big Data Applications and Services

Besides the three major themes above, key-terms cluster analysis also reveals that big data has a strong impact on every sector, from medical healthcare [24,25,63–65], finance and business [26,66–68], energy [69,70], government [71–74], education [75–77] to transportation [78–80]. The main source of big data comes from daily life and activities, for example, social media and networks. Facebook, as the most popular social media platform, has 2.4 billion users. It is followed by YouTube and WhatsApp, which have 2 billion and 1.6 billion users, respectively [81]. Embedded in the data generated by social media users is a rich collection of meaningful information and knowledge. With artificial intelligence, data science, and automation bots, these data can be used to improve marketing schemes and improve the online user experience. Another example is the application of big data in the health field. The healthcare ecosystem consists of many stakeholders, including patients, healthcare providers, pharmaceutical companies, medical device and service providers, IT solutions and service companies, health insurance, and government. With the explosion of clinical data, the issue of how to manage and process data from distributed and heterogeneous health IT systems data has become critical. Big data analytics systems make large-scale clinical data analytics possible and support the development of more efficient healthcare applications [56].

5.1.5. Big Data-Related Technologies

Based on key-terms cluster analysis, it can be seen that key technologies related to big data research that play an important role are cloud computing and the Internet of Things (IoT). Cloud computing provides resources and services that support big data processing and management, and IoT is one of the most significant sources of big data.

The definition of cloud computing according to National Institute of Standards and Technology is "a model for enabling ubiquitous, convenient, on-demand network access to a shared pool of configurable computing resources (e.g., networks, servers, storage, applications, and services) that can be rapidly provisioned and released with minimal management effort or service provider interaction". This cloud computing model consist of five main features, three service models, and four storage delivery models [82]. On-demand self-service, wide network access, resource combination, swift elasticity, and measured services are the main features of cloud computing. Furthermore, it offers three service models; i.e., software as a service (SaaS), platform as a service (PaaS), and infrastructure as a service (IaaS). The four cloud storage delivery models are public, private, hybrid, and community. Big data clouds refer to big data computing done over clouds [83].

Elasticity, pay per use, low investment and time to market, and transfer of risks are characteristics and major features of cloud technology that enable enterprises to build an integrated platform and elastically scalable infrastructure for deploying novel applications for big data analytics which were not economically feasible in traditional enterprise infrastructure [84]. Qubole and Altiscale are vendors that offer big data as a Service (BDaaS), in which enterprises can outsource the entire hardware, software, and infrastructure of big data. BDaaS systems encompasses the integration of PaaS and IaaS, PaaS and SaaS, or IaaS, PaaS and SaaS [85].

The Internet of things (IoT) refers to physical devices that have unique identities and are connected to the Internet [56]. IoT enables physical objects to exchange data and information with any connected device over the internet for monitoring or control functionality, meaning that the data will be transported over IoT networks all the time [86]. IoT is one of the triggers and sources of big data. The IoT system can utilize big data technology for data storage and analysis. The big data enabling technologies in the IoT context are related to data capturing devices, ubiquitous wireless communication, machine learning, and real-time analytics [87]. Xu et al. proposed an IoT big data analytics service known as time series analytics as a service on IoT (TSaaaS). This analytic service is used for performing pattern mining on a massive amount of collected sensor data and relies on the time-series database service and is accessible by a set of RESTful interfaces [88].

Cloud computing is a critical enabling technology that supports big data infrastructure, management, and processing. The topic of cloud computing has emerged since the initial phase of research on big data (2009–2015). Social media and networks are one source of big data and are the main research topic with distribution spread over most of the decade (2009–2015, 2017, 2018). The Internet of things (IoT) is also a source of big data, one that only appeared in the last phase (2018). Nevertheless, IoT is significant, and this topic appears in the analysis of the period (2009–2018), whereas social media and networks do not appear.

## 6. Further Discussions

### 6.1. Big Data Topic Analysis

Topic analysis is a text mining method based on semantic clustering [89]. Previous studies related to topic analysis in the field of big data are summarized and compared with the current study in Table 7.

**Table 7.** Review of topic analysis of big data research.

| Title and Reference | Objectives | Comments |
|---|---|---|
| Topic analysis and forecasting for science, technology and innovation: Methodology with a case study focusing on big data research [23] | To propose an analytic method for clustering associated terms and phrases to constitute meaningful technological topics and their interactions and identifying changing topical emphases. | The proposed method focuses on technology road mining (TRM). The results of the study are the identification of TRM components and TRM models that visualize objective results and qualitative discussions based on expert knowledge. |
| The major research themes of big data literature: From 2001 to 2016 [21] | To identify the major research themes in big data literature. | This paper clusters big data literature using the edge-betweenness clustering technique applied to the citation network. |
| Discovering and forecasting interactions in big data research: A learning-enhanced bibliometric study [22] | To discover interactions in big data research by detecting and visualizing its evolutionary pathways. | This paper integrates a machine learning and a bibliometrics approach to investigate global big data research from 2000 to 2015. The topic is defined based on the scientific evolutionary pathway model. |
| Research trends on big data in marketing: A text mining and topic modeling based literature analysis [26] | To identify the main trends on big data in marketing. | This study focuses on big data in marketing in an attempt to identify the trends in these applied domains using topic modeling methods. It provides a summarized overview of the literature by grouping articles in logical topics characterized by key relevant terms. |
| Theme mapping and bibliometrics analysis of one decade of big data research in the Scopus database [this paper] | To provide a comprehensive overview of big data research trends by integrating traditional bibliometrics and text mining approaches. | This study uses bibliometrics analysis and keyword clustering methods to systematically study and analyze the big data literature in the Scopus databases. Major research themes and future research directions are identified. |

*6.2. Future Big Data Research Trends*

In the following, the new trends, challenges, and opportunities in the future research of big data are discussed.

- Big data processing and analytics Driven by applications in the real world, big data analytics becomes very interesting and challenging. The need to process such extensive data becomes inevitable. However, as there is considerable noise in big data, cleaning and extracting the required data from such a colossal data space demand considerable effort. Furthermore, machine learning plays an essential role in transforming data into useful information and can be interpreted by humans. A vast amount of available data requires sophisticated machine learning algorithms and techniques. The challenge is that the process must take place in a distributed environment [90].
- Advanced supporting technologies for big data analytics Big data processing and analytics can only be implemented successfully when there is also innovation in the underlying supporting technologies. Efficient and scalable new algorithms for big data analysis need to be developed, as do tools and platforms.
- New or advanced big data applications and services In the future, the amount of data will continue to increase so that gigabytes will no longer be the relevant quantity of data. Billions of devices are connected to the internet and continuously emit data that must be collected and processed. Larger chunks of data, such as terabytes and petabytes, will be processed regularly. This rapid growth and processing of data have led to the emergence of entirely new applications for individuals and companies, such as remote healthcare systems and smart applications [90].

- Privacy and security in big data While the development of new big data technology enables better and more service automation, there are also privacy and security risks that come with this new technology that must be calculated and treated directly from the beginning. Big data technological advancements open up the possibility for more personal data to be collected and can cause severe privacy issues that must be addressed. Not only private data that needs to be secured but also public data and facilities connected to the internet can be targeted for attacks. Therefore, appropriate mechanisms must be developed and applied to reduce the risk of privacy and security.

*6.3. Threats to Validity*

This research is characterized by some data-related and study design limitations. First, the dataset used in this study originated solely from the Scopus database. Although Scopus is one of the largest international scientific databases, other reputable international databases, such as Web of Science (WoS) and Google Scholar, should be combined. Moreover, the selection of thematic fields was limited to three areas (computers, engineering, and mathematics), and only journal articles were chosen to ensure that research in progress was not part of the analysis. The second limitation concerns the use of keywords as clustering input. Although keywords chosen by authors or editors can be considered as representing the contents of a document, they may not adequately reflect the contents of an article. What the author wants to express is not necessarily summarized in the keywords and, thus, the results of the analysis can be misinterpreted. For future work, we recommend more in-depth content analysis with improvements based on the limitations above.

## 7. Conclusions

This study presents an overview of big data research landscapes based on the analysis of one decade (2009–2018) of journal publications from the Scopus database. Bibliometric and text mining analyses were conducted to obtain the following aspects of big data research: the growth of the output of journal publications, distribution of research productivity (countries, authors and institutions), publications of journals, the most popular articles, and major themes. This study reveals that research on big data is dominated by big data analytics, which includes methods, supporting infrastructure, and applications. It is crucial to understand that big data always comes together with analytics. Moreover, privacy and security issues are an essential aspect of big data research. Significant contributors to big data are social networks and the Internet of things, whereas the resources and services offered by cloud computing strongly support the management and processing of big data.

These results can be used as a reference for those who are interested in the development of big data research and fill the gap in the review of the literature. The methodology for identifying the major research themes applied in this study is unique and can also be adapted to other academic and technological fields.

**Author Contributions:** A.P. conducted the experiments and wrote the paper. H.M. and K.R. provided useful suggestions and corrections in the research and writing paper process. All authors have read and agreed to the published version of the manuscript.

**Funding:** This research was partly funded by the Ministry of Research, Technology, and Higher Education through the PDD scheme under contract number: NKB-1844/UN2.R3.1/HKP.05.00/2019.

**Conflicts of Interest:** The authors declare no conflicts of interest.

## Appendix A

**Table A1.** Top 10 characteristic keywords in each cluster for the time period 2009–2018.

| Labels | Keywords |
|---|---|
| 01 Cloud Computing | cloud computing, cloud, computer system, distribute, schedule, application, service, quality of service, resource, computing. |
| 02 Prediction | forecast, prediction, regression, time series, estimation, model, correlation, method, maximum likelihood, linear. |
| 03 Privacy | privacy, social, data privacy, data protection, personal, science, web, ethic, information, user. |
| 04 Image | image, feature extraction, feature selection, support vector machine, convolutional neural network, feature, detection, classification, recognition, deep learning. |
| 05 Internet of Things | internet of things, energy, monitor, intelligent, smart, sensor, cyber physical system, building, meter, transportation. |
| 06 Algorithm | cluster, algorithm, mapreduce, optimization, hadoop, k means, iterative, parallel, evolutionary, graph. |
| 07 Development | development, management, education, business, teaching, sustainable, industry, product, innovation, environmental. |
| 08 Memory | memory, storage, hardware, file, architecture, data storage, digital storage, random access, cache, query. |
| 09 Human | human, journal, priority, statistical, biology, procedure, genetic, genomics, article, study. |
| 10 Learning | learning, classification, learning system, machine learning, artificial intelligence, decision tree, on-line, data mining, e-learning, supervise. |
| 11 Security | security, wireless, communication, network, wireless communication, authentication, transmission, telecommunication, wireless network, protocol. |

## Appendix B

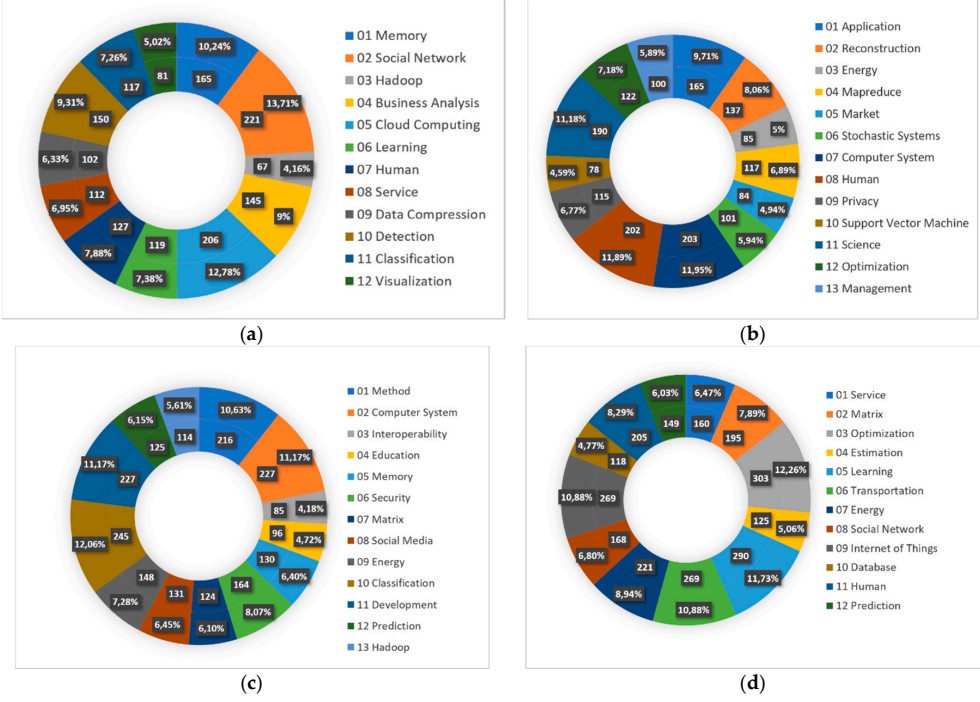

**Figure A1.** The number of keywords and their percentage, respectively, in each cluster. (**a**) 2009–2015; (**b**) 2016; (**c**) 2017; (**d**) 2018.

## Appendix C

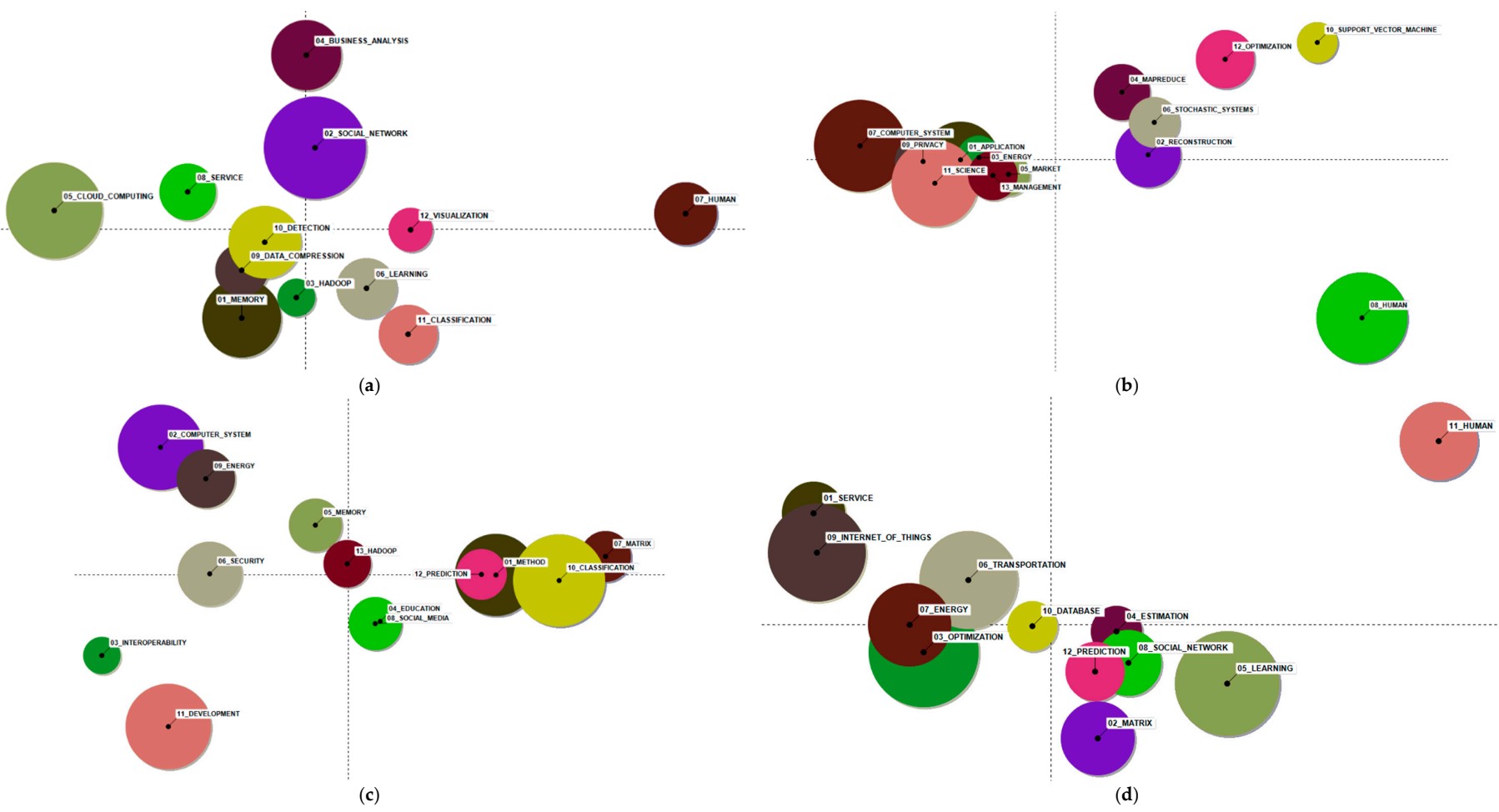

**Figure A2.** Keywords cluster visualization for each phase. (**a**) 2009–2015; (**b**) 2016; (**c**) 2017; (**d**) 2018.

## Appendix D

**Table A2.** Top 10 characteristic keywords in each cluster for the time period 2009–2015.

| Labels | Keywords |
|---|---|
| 01 Memory | query, parallel, memory, distribute, architecture, process, parallel processing, graphic processing unit, parallel programming, query languages. |
| 02 Social Network | social network, on-line, social media, web, social, identity, medium, twitter, community, influence. |
| 03 Hadoop | hadoop, mapreduce, k-means, jacobi, preference, job, cluster, iterative, apache, sample. |
| 04 Business Analysis | business analysis, government, privacy, science, information technology, competitive, society, business intelligence, business, research. |
| 05 Cloud Computing | cloud computing, cloud, internet of things, storage, security, mobile, key, application, computer system, access control. |
| 06 Learning | learning, learning system, machine learning, artificial intelligence, mixture, sparse, extreme learning machine, dictionary learning, knowledge, kernel. |
| 07 Human | human, informatics, factual, medical, genetic, procedure, biology, genomics, database, health. |
| 08 Service | service, management, manufacture, availability, information management, iplant, composition, production, chain, customization. |
| 09 Data Compression | schedule, data compression, heuristic, data flow, real time, compression, stream, road, bandwidth, reservation. |
| 10 Detection | detection, wireless sensor network, sensor, power, measurement, decomposition, smart grid, stability, communication, electric power. |
| 11 Classification | classification, classifier, support vector machine, decision tree, mining, frequent itemset mining, rule, association rule, feature selection, text |
| 12 Visualization | statistical, visualization, molecular, statistic, interface, visual, analysis, logit, symposium, multivariate. |

## Appendix E

**Table A3.** Top 10 characteristic keywords in each cluster for the time period 2016.

| Labels | Keywords |
|---|---|
| 01 Application | stream, application, computer, data handling, schedule, process, data processing, hardware, multimedia, memory |
| 02 Reconstruction | reconstruction, spatial, image, microscopy, regression, climate change, spatiotemporal, climate, temporal, action. |
| 03 Energy | energy, utilization, transmission, energy efficiency, consumption, allocation, power, wireless, electric, electric power. |
| 04 Mapreduce | cluster, mapreduce, k means, hadoop, algorithm, table, hadoop distributed file system, associative, keyword, header. |
| 05 Market | market, product, chain, knowledge, manufacture, supply, analytic hierarchy process, organizational, finance, internet. |
| 06 Stochastic Systems | stochastic, stochastic systems, text mining, factorization, text, information retrieval, pipeline, natural language processing, arabic, fabrication. |
| 07 Computer System | computer system, distribute, cloud computing, storage, network, data center, cloud, digital storage, radio, software defined network. |

**Table A3.** *Cont.*

| Labels | Keywords |
| --- | --- |
| 08 Human | human, journal, priority, study, factual, article, database, sequencing, information processing, procedure. |
| 09 Privacy | privacy, data privacy, security, encryption, cryptography, privacy preserving, data protection, secure, protection, personal |
| 10 Support Vector Machine | support vector machine, swarm, support vector regression, ridge, classification, feature selection, feature extraction, data mining, ensemble learning, synthesis. |
| 11 Science | science, web, research, technology, social, social network, internet of things, behavioral, smart city, on-line. |
| 12 Optimization | optimization, learning, forecast, prediction, kernel, method, collaborative filtering, algorithm, classification, recommendation system. |
| 13 Management | management, information management, waste, water, control, transportation, transit, nosql, manufacture, business. |

## Appendix F

**Table A4.** Top 10 characteristic keywords in each cluster for the time period 2017.

| Labels | Keywords |
| --- | --- |
| 01 Method | method, numerical, statistical, analysis, graphic, model, surface, graph, optimization, histogram. |
| 02 Computer System | computer system, distribute, internet of things, wireless sensor network, sensor, real time, cloud computing, embedded system, smart city, routing. |
| 03 Interoperability | interoperability, emission, information system, aesthetic, semantic, customer, bold, disaster, semantic web, interview. |
| 04 Education | education, convolutional neural network, teaching, student, image, convolution, facial, deep belief network, deep learning, recognition. |
| 05 Memory | memory, storage, hardware, data storage, random access, reconfigurable, file, architecture, digital storage, flash. |
| 06 Security | security, privacy, data privacy, cryptography, privacy preserving, encryption, protection, data protection, encryptions, product. |
| 07 Matrix | matrix, algebra, subspace, decomposition, cluster, discriminant, density, principal component analysis, sparse, low rank. |
| 08 Social Media | social media, biology, human, social network, twitter, genetic, informatics, vaccine, disease, on-line. |
| 09 Energy | energy, utilization, energy efficiency, consumption, virtual machine, mobile, power management, transmission, meter, wireless network. |
| 10 Classification | classification, learning system, learning, machine learning, artificial intelligence, decision tree, algorithm, extreme learning machine, tree, rough. |
| 11 Development | development, management, sustainable, innovation, environmental, economic, society, technology, education, supply. |
| 12 Prediction | prediction, forecast, inference, wind, recommendation system, fuzzy, particle swarm optimization, markov chain monte carlo, cluster, fault |
| 13 Hadoop | hadoop, mapreduce, hadoop distributed file system, nosql, query, hbase, mongodb, rock, data warehouse, software. |

## Appendix G

**Table A5.** Top 10 characteristic keywords in each cluster for 2018.

| Labels | Keywords |
|---|---|
| 01 Service | service, cloud, cloud computing, provider, storage, product lifecycle, mobile, healthcare, edge computing, digital. |
| 02 Matrix | cluster, matrix, mining, algorithm, iterative, association rule, rough set, recommendation system, approximation, data mining. |
| 03 Optimization | schedule, distribute, optimization, program, parallel, high performance computing, computer system, application, resource allocation, memory. |
| 04 Estimation | estimation, statistic, travel, priority, covariance, journal, test, wasserstein, hazard, seismic. |
| 05 Learning | learning, machine learning, classification, learning system, artificial intelligence, supervise, algorithm, e-learning, imbalanced, semi supervised. |
| 06 Transportation | transportation, road, transmission, network, privacy, communication, vehicle, radio, wireless, system. |
| 07 Energy | energy, sustainable, energy efficiency, consumption, utilization, management, conservation, supply, environmental, business analysis |
| 08 Social Network | social network, social media, detection, on-line, extraction, twitter, fault, feature extraction, sentiment analysis, classification. |
| 09 Internet of Things | internet of things, smart, wireless sensor network, smart city, security, sensor, authentication, technology, encryption, cryptography |
| 10 Database | query, database, cache, nosql, transaction, process, mongodb, query languages, punch, tunnel. |
| 11 Human | human, image, medical, procedure, disease, adult, record, drug, image processing, male. |
| 12 Prediction | forecast, prediction, regression, time series, arima, short term, precipitation, deep belief network, stock, convolutional neural network. |

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
