# Peer review of "Theme Mapping and Bibliometrics Analysis of One Decade of Big Data Research in the Scopus Database"

_information, doi:10.3390/info11020069_

Round 1

Reviewer 1 Report

I believe that the paper analyzes a current and interesting topic: the evolution of big data publications.
Although traditional bibliometric techniques are applied, I consider that the work is well estructured out and the methodological development seems correct.
However, it is suggested that the authors justify in greater detail why only 3 thematic fields have been selected (Computer Science, Engineering, Mathematics) to study Big Data production (are there any previous studies that allow us to conclude that activity on Big Data in other fields is not relevant?).
With respect to the results obtained, it is observed that the production of countries and institutions is obviously affected by size. To reduce this bias, it is suggested that the authors use some index of normalization (perhaps the activity index) to show which are the most specialized countries and institutions and to complement the information provided.
With respect to the results in which the production by author appears, it would be interesting to go a little deeper (in the methodological section) to understand how the normalization and disambiguation of names has been carried out.
Finally, it is suggested that the authors expand the discussion section by including comparisons with previous results (both in the Big Data analysis and in the methodological development) and explain more explicitly what the contribution of the present study is.
Including the suggested changes, I believe that the paper is suitable for publication in the journal.

Author Response

Dear Editors and Reviewers

MDPI Information

We are pleased to submit our revised manuscript entitled “Theme Mapping and Bibliometrics Analysis of One Decade of Big Data Research in the Scopus Database,” authored by Anne Parlina, Hendri Murfi, and Kalamullah Ramli.

We appreciate the time and effort that the editor and referees devoted to reviewing this manuscript. Please find our response to reviewers’ comments and suggestions in the attachment.

To make the revisions easy to locate for the editors and reviewers, we have marked the changes in the manuscript using the “track changes” and “comments” function in Microsoft Word.

We hope the reviewers agree to reconsider our revised version for publication in Information.

Sincerely yours,

Anne Parlina

Reviewer 2 Report

Some words in Abstract are way too abstract. For example: 

'Other critical 22 aspects of big data research are security and privacy, while the technologies which are closely 23 related to big data are the internet of things and cloud computing.'

 -- Closely related in what context, enabling technologies? technologies supported by Big Data? 

The paper needs careful proof-reading to correct language and grammatical issues.

One of the fundamental question is that in the presence of well established methods like Systematic Mapping Study, Systematic Literature Review, why authors use a different mechanism for literary analysis? The justifications needs to be clear and explicit.

Why only scopus is used? why not other relevant digital libraries like IEEE Xplore, ACM, etc.

Figure 1 needs to be fixed it has red/blue underlines placed by MS Word.

Section 3 is too large, it needs to be broken down into multiple sections.

Section 3.3.4 is way too small.

An explicit Section of Future Research Trends is missing.

Overall the paper is well written and easy to follow. It needs following improvements.

 - Language and Grammar Check

 - Why Systematic Mapping Study/Systematic Literature Review metod is not being adopted?

 - Why only SCOPUS?

 - Better organisation of the paper into Sections. Section 3 is way too large.

 - Future Directions need to be mentioned explicitly.

- Threats to validity must be mentioned.

A section is needed between Introduction and Method that discusses and overviews the details of big data systems, preferably with a Figure.

Author Response

(The authors gave the same response as above.)

Round 2

Reviewer 2 Report

The authors have made significant efforts to revise the manuscript and address the previously raised comments. Based on the revised manuscript, I would suggest to accept the updated version of the manuscript.

Author Response

Dear Reviewer 2,

Thank you very much for reviewing our manuscript. We also greatly appreciate you for your complimentary comments and suggestions. The manuscript has certainly benefited from these insightful revision suggestions.

Thank you again for consideration of our revised manuscript.

Best regards,

Anne Parlina

Department of Electrical and Computer Engineering
Faculty of Engineering - Universitas Indonesia
Depok 16424 – Indonesia
Email: [email protected]